# Carbon Nanocomposite Membrane Electrolytes for Direct Methanol Fuel Cells—A Concise Review

**DOI:** 10.3390/nano9091292

**Published:** 2019-09-10

**Authors:** Gutru Rambabu, Santoshkumar D. Bhat, Filipe M. L. Figueiredo

**Affiliations:** 1CICECO—Aveiro Institute of Materials, University of Aveiro, 3810-193 Aveiro, Portugal; 2CSIR—Central Electrochemical Research Institute-Madras Unit, CSIR Madras Complex, Chennai 600 113, India

**Keywords:** carbon nanotubes, graphene oxide, proton exchange membranes, direct methanol fuel cell

## Abstract

A membrane electrolyte that restricts the methanol cross-over while retaining proton conductivity is essential for better electrochemical selectivity in direct methanol fuel cells (DMFCs). Extensive research carried out to explore numerous blends and composites for application as polymer electrolyte membranes (PEMs) revealed promising electrochemical selectivity in DMFCs of carbon nanomaterial-based polymer composites. The present review covers important literature on different carbon nanomaterial-based PEMs reported during the last decade. The review emphasises the proton conductivity and methanol permeability of nanocomposite membranes with carbon nanotubes, graphene oxide and fullerene as additives, assessing critically the impact of each type of filler on those properties.

## 1. Introduction

Direct methanol fuel cells (DMFCs) belong to a category of polymer electrolyte membrane fuel cells (PEMFCs) that utilize a liquid fuel to produce electrical power at room temperature. DMFCs do not require humidification accessories which significantly reduce the complexity, the volume and weight of the system compared to H_2_-fueled PEMFCs. In addition, handling of liquid fuels, such as methanol, is easy in comparison with the handling of hydrogen gas [1,2,3].

The transport sector contributes to most of the atmospheric pollution and consumes a major portion of the energy generated worldwide. On the other hand, the portable electronic market (e.g., smartphones and laptops) is growing rapidly and the state-of-the-art Li batteries are lagging in some aspects, including safety. A portable DMFC system is expected to provide solutions to these mobile electronics, hence the development of an ideal DMFC system is of prime importance in the current fuel cell research. A DMFC comprises several components, namely the membrane electrode assembly (MEA), the flow channels, endplates and current collectors. The MEA is the key component comprising a polymer electrolyte membrane sandwiched between an anode and a cathode. The state-of-the-art PEM is a perfluorosulfonic acid (PFSA) membrane (of which Nafion^®^ is the most well-known commercial example). The anode material is usually a Pt-Ru bimetallic catalyst supported on carbon, while the cathode is Pt supported on carbon [4].

In the typical DMFC system schematized in Figure 1, aqueous methanol is fed to the anode side of the MEA where the electrochemical oxidation of methanol generates protons, electrons and carbon dioxide according to CH_3_OH + H_2_O → CO_2_ + 6H^+^ + 6e^−^, with reversible potential E^0^ = 0.046 V. The protons at the anode pass through the solid polymer electrolyte to the cathode, where they combine with the electrons arriving through the external circuit and the oxygen 3/2O_2_ + 6H^+^ + 6e^−^ → 3H_2_O, E^0^ = 1.23 V.

The ideal MEA materials have pre-requisites as follows:Efficient anode catalyst for complete electro-oxidation of methanol.Solid polymer electrolyte with high proton conductivity and low methanol permeability.Methanol-tolerant cathode catalyst with high oxygen reduction activity.

Among all MEA components, the PEM plays major multiple roles acting as a physical barrier between anode and cathode to avoid mixing of fuel and oxidant, an insulator for electrons, and, finally, as an electrolyte ensuring the selective transport of protons from anode to cathode. For practical DMFC applications, the PEM is required to have high proton conductivity over a wide range of temperatures, low fuel permeation, good chemical, mechanical and thermal properties, along with durability.

As stated above, Nafion^®^ membrane is used as a current state-of-the art PEM for DMFCs for its remarkable mechanical and chemical stability and high proton conductivity. Nafion^®^ consists of a strong hydrophobic fluorinated backbone and hydrophilic pendant sulfonic acid chain, as shown in Figure 2.

The backbone offers remarkable mechanical strength along with chemical stability, and the sulfonic acid groups improve the water retention capacity and are responsible for its superior proton conductivity [5]. However, the aqueous domains formed in the vicinity of these ionic clusters also lead to high methanol permeation from the anode to the cathode, where it is oxidized creating a mixed potential that reduces the overall efficiency of the cell [6,7]. Hence research efforts made to overcome the above issues follow two main different approaches viz. modification of PFSA membrane with organic/inorganic additives, and development of alternative polymeric composites.

The dispersion of a variety of inorganic additives like silica, zirconia, metal oxides and zeolites to form Nafion^®^ composites is widely suggested in the literature as a mean to decrease the methanol permeability in PFSA membranes [8,9,10,11]. These composite membranes show restricted methanol permeability with a compromise of proton conductivity. On the other hand, many organic additives have been explored to form composites, blends and cross-linked membranes of Nafion^®^ [12,13,14,15] with better proton conductivity.

Other PEMs based on aromatic hydrocarbon polymers aliphatic polymers and their composites are explored as suitable electrolytes in DMFC [16,17,18,19,20,21,22]. Among all the sulfonated aromatic polymers, polyether ether ketone (PEEK), polyether sulfone (PES), polyether nitrile (PEN) and poly imides (PI) shown in Figure 3 are considered as the most suitable for DMFC due to their superior thermal, mechanical and chemical stability, good film-forming ability and extremely low fuel permeability and cost compared to Nafion^®^ [23]. These aromatic polymers are generally sulfonated using electrophilic substitution or by polymerizing the monomers having sulfonic acid groups to implant proton conducting groups. Usually, a high degree of sulfonation (DS) leads to high proton conductivity. However, its mechanical stability is affected badly due to excessive swelling at such a high DS, thus limiting their direct use as PEMs. Numerous efforts have been made to prepare composites of these sulfonated polymers using a variety of micro/nano-additives such as inorganic oxides. Although these composite membranes show significant development in terms of methanol crossover, they have lower proton transport and further improvement is needed to use them as PEMs for DMFC, hence the fabrication of aromatic sulfonated polymer composites has been the subject of continued study.

Aliphatic polymers such as polyvinyl alcohol (PVA) are another option for exploring the composite membranes for DMFC. PVA is a semi-crystalline polymer well studied in the area of water/methanol separation. PVA has good methanol resistance which is a key point for DMFC electrolyte. However, it is soluble in water due to its hydrophilicity, hence it should be stabilized using cross-linking agents such as glutaraldehyde (GA) (Figure 4).

Cross-linking of hydroxyl groups reduces the swelling and improves the stability of the membranes, but the mechanical properties and the proton conductivity of PVA needs considerable improvement, hence the fabrication of PVA-carbon composites are explored to make it more suitable electrolyte for DMFCs.

## 2. Carbon Nanomaterials as Additives in PEMs

Carbon nanomaterials, like CNTs, graphene and fullerene, were explored as reinforcing additive materials in PEMs due to their excellent mechanical stability and methanol-blocking characteristics. The high electronic conductivity of these materials is the bottleneck for their applications in PEMs, where it is detrimental. If the carbon filler fraction is too high, the level of electronic conductivity increases to the point where the membrane becomes an electronic conductor and loses its function as a PEM. For example, the literature reports 2–3 wt.% of CNTs in PEMs as the percolation threshold leading to excessive electronic conductivity of the membrane [24]. However, it varies depending upon the polymer matrix, intrinsic electrical properties of host matrix and structural orientation in the polymer matrix [25]. Another key point is their dispersion in the polymer matrix since heterogeneously-dispersed fillers often lead to mechanical instability and poor transport properties. Hence, the surface functionalization of the filers is employed to achieve their homogeneous distribution, wherein interfacial interactions between the functional groups of filler and polymer play a vital role.

There are few recent review articles centred on the use of carbonaceous materials in fuel cells, with particular emphasis on GO [26,27,28]. For example, the attempt of You et al. to review the application of CNTs, mesoporous carbon, carbon black, carbon nanofibers, aerogel, nanocoils, graphene and fullerene as additives in electrodes and membranes for fuel cells is necessarily too general [26]. On the contrary, the Panday et al. [27] and Farooqui et al. [28] reviews are restricted to the application of GO fillers.

With a very different perspective, our text presents a comparative analysis of various kinds of nanocarbon-based fillers for polymer electrolyte membranes for application in DMFC, emphasizing their effect on the proton conductivity, methanol permeability and mechanical stability of the membranes. Such comprehensive coverage is not found in the literature until this date. The review is based on an original and extensive selection of literature work on carbon nanomaterial composite membranes listed, which is given in Table 1.

Among the carbon materials, graphene oxide and carbon nanotubes have been given greater attention according to the analysis of bibliometric data from last ten years depicted in Figure 5. This justifies the relative coverage given to each type of filler in the following sections.

### 2.1. Carbon Nanotubes as Additives for PEMs in DMFC

CNTs are considered to be promising additive material due to their unique structural and physical properties. CNTs form tubular structures with a diameter of nanometre-scale and length in the micrometre range. CNTs exist in different forms namely single-walled carbon nanotubes (SWCNTs) and multi-walled carbon nanotubes (MWCNT) [82]. SWCNTs are more ordered structures and with superior flexibility and electronic conductivity than MWCNTS. As far as PEMs are concerned, MWCNTs are preferred over SWCNTs due to lower electronic conductivity and high surface defects, which are essential for surface modification. The tensile strength of CNTs is around 63 GPa, which is 50 fold higher than steel, while Young’s modulus is five-fold higher [82]. In addition to these remarkable mechanical properties, high surface area boost CNTs as reinforcing material in polymer matrices [34,44,83] CNTs are chosen as additive mainly to address the methanol permeability and mechanical strength issues of PEMs in DMFCs. However, the homogeneous dispersion of CNTs is difficult as they are held by Van der Waals forces, which limit the interfacial interactions with the polymer matrix [84]. To tackle these issues, surface functionalization of CNTs has been performed with various functional groups such as silica or chitosan, but mostly acids (sulfonic acid, phosphonic acid, carboxylic acid and heteropoly acids) [24,36,84,85]. Figure 6 schematizes these different approaches for the surface functionalization, which are detailed in the following subsections.

#### 2.1.1. PFSA-CNT Composite PEMs

Among PFSA ionomers, Nafion^®^ is the most explored PEM in DMFC. Composite PEMs of Nafion^®^ fabricated using functionalized CNTs solve one of its most important drawback, which is the high methanol permeability.

Thomassin et al. used CNTs functionalized with carboxylic acid to prepare Nafion^®^ composites by melt-extrusion process [30]. These –COOH-CNTs could decrease the methanol permeability up to 60% and improve the mechanical stability. On the other hand, the ionic conductivity could not be improved much as these carboxylic acids are less acidic than the sulfonic acid groups of Nafion^®^ and thus, the dissociation of protons is difficult. However, the overall selectivity of the Nafion^®^-COOH-CNT membrane is improved by a factor of 2, which is comparable or better than the results obtained for other composite membranes containing, e.g., silica particles (a 1.5 times increase) [86,87], a combination of zirconium oxide and zirconium phosphate (a 2.3 times increase), [88] and palladium nanoparticles (a 1.5 times increase) [89].

Proton conduction in Nafion^®^ is assisted by water molecules solvating the sulfonic acid groups, forming well-connected proton-conducting aqueous ionic domains, hence the membrane must be sufficiently humidified in order to achieve appropriate proton conduction levels. This leads to limited temperature operation (<80 °C) wherein CO poisoning occurs reducing the utilization of the Pt catalyst. In this context, materials containing imidazole, such as ionic liquids, are explored for fuel cell applications as they contain nitrogen groups, which are capable of forming hydrogen bonding networks to assist proton conduction even under dehydrated conditions. Imidazole-functionalized materials are appropriate to use as reinforcing additives in PEMs as they localize proton carriers to form well-defined ionic clusters (Figure 7). In this regard, MWCNTs functionalized with imidazole can be used as a nano-additive to Nafion^®^ [29]. The homogeneous dispersion of these Im-CNTs in Nafion^®^ can be explained on the basis of charge repulsions wherein imidazole grafting reduces the negative charge of –COOH-CNTs and convert CNTs into a positively-charged forming stable dispersion. The imidazole group promotes the proton transport via Grötthus type mechanism by forming the hydrogen bonding network between the nitrogen moieties and the sulfonic acid groups. This is beneficial for high temperature operation of Nafion^®^ membrane, On the other hand, a significant decrease in methanol permeability is also observed for Nafion^®^-Im-MWCNTs composites due to the change in nano-channel formation by the interactions between basic moiety of Im-CNTs and sulfonate groups of Nafion^®^.

Chitosan-functionalized CNTs have also been explored to prepare Nafion^®^ composite membranes [31]. Chitosan, a natural linear polysaccharide (cationic), can interact non-covalently with CNTs forming a stable dispersion while the hydrophobic groups of chitosan improve the interfacial adhesion of CNTs [90]. When these chitosan-modified CNTs are dispersed in Nafion^®^, the cationic part of chitosan interact with the anionic part of Nafion^®^ forming well dispersed Nafion^®^-CNTs composites. Moreover, the electrostatic interactions and hydrogen bonding between carboxyl functional groups of chitosan and sulfonic acid groups of Nafion^®^ constitute a percolated path for proton conduction. This explains the enhanced conductivity of a Nafion^®^ membrane loaded with 0.5 wt.% chi-CNTs in comparison to the pure matrix (104 mS cm^−1^ vs. 86 mS cm^−1^), whereas the same loading of unmodified CNTs lowers the conductivity (to 73 mS cm^−1^). Chitosan also leads to a reorganization of the Nafion^®^ nanomorphology reducing the size of the aqueous domains, which has direct effect on methanol transport and the permeability of these membranes up to one order lower than the pristine Nafion^®^. The durability of the membranes assessed in 5 M methanol solution by static (OCV) and dynamic (load of 200 mA cm^−2^) methods did not show an obvious drop in performance during ca. 100 h. However, further investigation on lifetime of these composite membranes are essential as these test conditions are indeed insufficient.

The modification of the surface of CNTs with silica (SiO_2_) nanoparticles is another approach to successfully disperse CNTs in Nafion^®^ and enhance its properties [32]. In spite of substantial reduction in methanol permeability (from 2.25 × 10^−^^6^ cm^2^ s^−1^ to 3.12 × 10^−7^ cm^2^ s^−1^), Nafion^®^-SiO_2_-CNTs composite membrane shows lower proton conductivity due to the disturbance in supramolecular self-assembly of Nafion^®^ [91]. To eliminate this problem, SiO_2_-CNTs further immobilized with phosphotungstic acid (PWA) can actively participate in proton transport by interacting with hydrophilic sulfonic acid groups of Nafion^®^. The conductivity of Nafion^®^-PWA-SiO_2_ composite is maintained on par with pristine Nafion^®^, while the methanol permeability is further reduced (to 2.63 × 10^−7^ cm^2^ s^−1^).

#### 2.1.2. Non-Fluorinated Polymer-CNTs Composites

Among non-fluorinated polymer, sulfonated polyether ether ketone (SPEEK) is relatively more explored as PEM in DMFCs. Many attempts have been made to prepare composites of SPEEK for DMFC using functionalized CNTs as reinforcing additives. Li Cui et al. fabricated composite poly electrolytes of SPEEK using silica-coated CNTs [43]. The hydrophilic SiO_2_ layers on CNTs improve the dispersion as well as the insulating characteristics of SiO_2_ minimize the risk of short-circuiting even at higher loadings (5 wt.%). The composite membranes with SiO_2_-CNTs improve water uptake while reducing the swelling in comparison to neat SPEEK membrane. The hygroscopic nature of SiO_2_ coated layer on CNTs lead to improved water retention while the robust characteristics of CNTs restrict the chain flexibility of SPEEK improving the dimensional stability. The results obtained for SPEEK-SiO_2_-CNTs are similar to those obtained for Nafion^®^-SiO_2_-CNTs [32]. A significant decrement in proton conductivity up to 50% is observed for 5 wt.% SPEEK-SiO_2_-CNTs due to reduced sulfonic acid concentration, while the methanol permeability is lowered by one order of magnitude (3.42 × 10^−7^ to 4.22 × 10^−8^ cm^2^ s^−1^). This is attributed to the occupation of SiO_2_-CNTs in hydrophilic channels of the PEM improving the tortuosity for methanol diffusion.

To sort out the issues related to low proton conductivity in this kind of composites, CNTs can be functionalized with proton conducting functional groups such as sulfonic acid and phosphonic acid [24,44]. These functional groups play a vital role in the proton conduction of on CNTs by the reorganization of hydrogen bonds between the proton hopping sites. In the composite membranes, the CNTs treated with sulfonic (S-CNTs) and phosphonic (P-CNTs) acid trigger a uniform distribution of ionic clusters, which are relatively smaller than that of neat polymeric membrane. The uniform and well-connected ionic clusters promote proton transfer up to 50% with S-CNTs and 166% with P-CNTs, while the smaller ionic clusters offer a torturous path, thus reducing methanol permeability to 77% for S-CNTs and 40% for P-CNTs from its original value. Our recent study explored the use of CNTs grafted with polystyrene sulfonic acid (PSSA) as an additive in SPEEK matrix to form composite membranes [40]. PSSA is an amphiphilic polymer that will cause less damage to the tubular structure of CNTs in comparison to strong acid treatments such as H_2_SO_4_. In addition, the hydrophobic part of styrene form π-π interactions with CNTs, while the hydrophilic sulfonic acid groups impart hydrophilicity leading to improved dispersion of CNTs by counteracting Van der Waals interactions between the tubes. Incorporation of PSSA-CNTs in SPEEK and sulfonated poly(arylene sulfone) (SPAS) has significant improvement in mechanical strength up to 43% and 460% respectively due to superior mechanical properties of CNTs [36,40]. However, they improve proton conductivity only by 28% for SPEEK and 13% for SPAS. These enhancements are low compared to other sulfonated CNTs composite membranes, which may be due to lower functional group density on CNTs. Pt-Ru decorated CNTs dispersed in SPAS matrix can also improve the performance of DMFC because Pt-Ru helps to oxidize the methanol within the membrane while passing through it, thus preventing methanol crossover to the cathode. The lifetime analysis of these composites, also performed by recording OCV as a function of time (up to four days), revealed improved OCV of the Pt-Ru-CNTs composites, whereas the neat SPAS membrane shows a drop in OCV due to severe methanol crossover [36].

The orientation of CNTs in polymer matrices has a significant effect on membrane performance. CNTs can be oriented by an applied electric or magnetic fields [92,93,94]. Swati Gahlot et al. prepared composite membranes of SPEEK with electrically aligned CNTs functionalized with carboxylic and sulfonic acid groups [41]. The electric field applied to the SPEEK/CNT mixture in dimethylacetamide introduces a dipole moment due to the variation of the dielectric constant of SPEEK and CNTs, thus leading to the alignment of the CNTs in the direction of the applied electric field (Figure 8). The vertical alignment of tubes in the membrane together with their functional groups contributes to enhanced proton conductivity up to 16% for C-CNTs and 88% for S-CNTs in comparison to pristine SPEEK. These values are also ca. 30% higher than for randomly aligned S-CNTs, which is ascribed to the higher water uptake in the pores created by the vertical alignment of S-CNTs in SPEEK.

#### 2.1.3. PVA-CNT Composite PEMs

PVA is one of the most explored low-cost PEMs, but the poor mechanical properties along with low conductivity hinder many practical applications, namely DMFCs. To overcome these issues functionalized CNTs such as S-CNTs or phosphotungstic acid-CNTs (PWA-CNTs) were used as reinforcing materials in PVA. Two different methods have been suggested to sulfonate CNTs: (1) by thermal decomposition of ammonium sulphate [38]; and (2) chemical functionalization by amino methane sulfonic acid [37]. These two types of CNTs show distinct behaviour when incorporated in PVA. The differences are explained on the basis of concentration of sulfonic acid functional groups anchored on CNTs, which is much higher in the case of the chemical functionalization with amino methane sulfonic acid. This is also reflected in the conductivity results, with the ammonium sulphate approach leading to a marginal increment in conductivity, and the amino methane sulfonic acid increasing the conductivity by almost 50% compared to neat PVA. On the other hand, both types of CNTs contribute to reduced methanol permeability. Yinhui Li et al. reported immobilization of PWA on CNTs using poly(diallyldimethylammonium chloride) (PDDA) to form PDDA-PWA-CNTs and is used as a filler to PVA membranes [39]. It is reported that 2 wt.% of these PDDA-PWA-CNTs fillers improve the mechanical strength of PVA for about 270% while the swelling of the membrane is reduced by 50%. On the other hand, the proton conductivity was improved from 4 mS cm^−1^ to 9.4 mS cm^−1^ due to well-distributed PDDA-PWA-CNTs forming a hydrated layer for facile proton transport.

### 2.2. Graphene Oxide as an Additive for PEMs in DMFCs

Another class of carbon material used as filler in PEMs is graphene oxide (GO). GO can be derived by the exfoliation of graphite and subsequent oxidation. Graphene oxide contains carboxylic acid groups and is readily dispersible in many solvents due to the hydrophilicity imparted by the oxygen-rich functional groups [95,96,97]. GO has a thickness of a single atomic layer while the lateral dimensions are in the order of few microns. The presence of carboxyl and hydroxyl functional groups of GO and the ease of further functionalization with proton conducting groups like sulfonic acid, which facilitate the proton transport is an added advantage. GO is known as an electronic insulator while its proton conductivity is as high as 10^−2^ S cm^−1^, which makes GO as an attractive filler for PEMs [98]. GO is further functionalized to improve the properties and to use as nano-filler in PEMs, some of the most common functionalization methods of GO are shown in Figure 9.

Thanks to large surface area and electronic insulation properties of GO which enables its use as nano-filler in ionomer membranes. The flexibility and compatibility of GO with the host polymer matrix leads to significant improvement in the mechanical properties of the membranes [99]. Due to the methanol impermeable nature of GO, it could effectively arrest the methanol crossover when used as an additive in PEM. Unlike most of the inorganic material, GO can restrict the methanol crossover without much compromise in proton conductivity as it has carboxylic acid groups and hydroxyl groups which act as proton conductors.

#### 2.2.1. PFSA-GO Composites

GO and its derivatives are identified as potential additives to control methanol crossover of PFSA membranes such as Nafion^®^. There are many reports on the methods of fabrication of Nafion^®^ composites using GO and its derivatives, but most of them are based on the lab-scale solution cast technique. Composite membranes prepared by this method using GO often show reduced methanol crossover. The interactions between GO and Nafion^®^ can be explained on the basis of their amphiphilic nature, GO can interact with both hydrophobic backbone and pendent hydrophilic sulfonic acid groups thus providing better compatibility and homogeneous distribution [47,95]. Incorporation of GO in Nafion^®^ reduces the methanol permeability with a slight compromise in proton conductivity. However, the overall selectivity of the composite membranes is higher than pristine Nafion^®^. The reduction in methanol permeability is attributed to the reduction of ionic cluster size of Nafion^®^ with the incorporation of GO [47]. The restricted methanol permeability for the composite membranes enables them to operate at higher methanol concentration. For example, for 5 M methanol solution, methanol permeability is predominantly higher than 1 M, but even at 5 M methanol, the composite membrane shows peak power density of 71 mW cm^−2^ whereas Nafion^®^-112 shows a peak power density of 26 mW cm^−2^ [47].

The use of sulfonated GO (S-GO) fillers in Nafion^®^ was also reported with the aim to minimize the conductivity loss observed when using pure GO fillers [46]. This was achieved by treating GO with a mixture of nitric acid and sulphuric acid using microwave, and then incorporating these particles into the Nafion^®^ matrix. It is apparent that, as for pure GO, S-GO also decreases the average dimension of ionic cluster size, thus decreasing methanol transport. Proton transport, on the contrary, is improved for the Nafion^®^-S-GO composite due to significant increment in water content bound to the sulfonic acid groups. Another benefit from the S-GO filler, analogous to other fillers [30,100,101,102], is the improvement of the dynamic mechanical storage modulus of the Nafion^®^-SGO composite membrane in comparison with pristine Nafion^®^ [103]. This enhanced mechanical stability enables the use of thinner membranes, which have the advantage of lower resistance in fuel cell operation. The dynamic mechanical analysis of these composite membranes also indicates the onset of long-range mobility of the polymeric chains (the α-relaxation) at a temperature higher than the pristine membrane. The storage modulus was 1.9 times higher and a 20 °C higher tan δ peak was observed for Nafion^®^-SGO composites in comparison with pristine Nafion^®^ membrane due to the large interfacial area between S-GO and Nafion^®^. The authors also made fuel cell tests of a composite membrane with 0.05 wt.% of S-GO in Nafion^®^, which yielded a 42 mW cm^−2^ at 0.4 V, whereas Nafion^®^-115 shows 32 mW cm^−2^ under similar conditions [103].

The change in the bi-continuous micro-structure was observed in Nafion^®^ loaded with S-GO-SiO_2_ due to the compatibility of GO and the sulfonic acid groups of SGO-SiO_2_ [48]. It is noteworthy that at 0.5 wt.% loading of S-GO-SiO_2_ in Nafion^®^, the microstructure of Nafion^®^ is aligned in non-uniform fashion due to the interaction of sulfonic acid groups present in the nano-filler and polymer. At higher loadings (0.8 wt.%), the compatibility of GO is dominant wherein aggregation of S-GO-SiO_2_ lowers the concentration of sulfonic acid groups. In contrast to the observation on the simpler Nafion^®^-S-GO composites, [47] the water uptake of Nafion^®^-S-GO-SiO_2_ membranes is greatly improved compared to pristine Nafion^®^ membrane due to the hygroscopic nature of SiO_2_. The Increased water content in the membrane is suggested to explain the enhanced protonic conductivity of the Nafion^®^-SGO-SiO_2_ composites. In addition, the ionic conductivity is influenced by the dimension and connectivity between the sulfonic acid groups [104,105,106]. SGO-SiO_2_ improves the connection between sulfonic acid groups thus providing continues path for proton conduction while GO sheets act as methanol barrier, Figure 10 shows the schematic representation of Nafion-SGO-SiO_2_ membrane illustrating proton and methanol transport.

On the other hand, membrane preparation method is also greatly influence the properties of composite membranes. Casting on Petri dish is a lab-scale preparation method of such kind of composite membranes. Up-scaling implies significantly different processing methodologies, which may invalidate much of the conclusions drawn from simply casted membranes. The casting procedure and removal of solvent has a significant effect on membrane properties via local reorganization of ionic domains [45]. Nafion^®^-functionalized GO composite membranes prepared in two different casting methods, i.e., solution cast on a Petri dish and casting using doctor blade has shown considerable differences in their microstructures producing a drastic effect on solvent mobility (water and methanol) in the membrane [107]. In the membranes prepared by casting on Petri dish, GO layers are oriented orthogonally to the membrane surface and are capable of coordinating linear superstructures in the hydrophobic domains. Whereas in the membranes prepared using a doctor’s blade, GO layers are preferentially oriented parallel to the membrane surface. Due to this distinct orientation of nano-fillers, the membrane exhibit a large difference in their water/methanol diffusion coefficients. The nano-composite membranes prepared on the Petri dish show 50%/53% of water/methanol uptake while these figures become 27%/31% for doctor blade method. On the other hand, the difference is negligible for recast Nafion^®^ membranes prepared in both the methods. The maximum power density was observed for the membrane prepared in a Petri dish wherein water diffusion and retention of this membrane offers lower resistance producing high current density. At higher temperatures (>100 °C) higher tortuosity effects was observed for the membranes prepared using a doctor’s blade, offering lower methanol crossover, hence depending upon operating conditions of DMFC these preparative methods (Petri dish/doctor blade) can be chosen for better performance.

The other methods to prepare Nafion^®^-GO composites is lamination of GO paper on commercial Nafion^®^ membrane through transfer printing followed by hot press [72]. The thickness of the GO paper can be optimized based on the requirements. GO laminated Nafion 115 membrane (Figure 11) show 70% depression in methanol permeability and a decrement of 22% in proton conductivity. These membranes are effective in arresting the methanol transport compared to the membranes prepared by solution cast method. However, laminated membranes show a larger reduction in proton conductivity compared to GO incorporated membranes due to the internal resistance between the layers.

#### 2.2.2. SPEEK-GO Composite PEMs

As stated above, SPEEK is considered as best alternative polymer to Nafion^®^ due to its reduced methanol permeability and the presence of narrow and more branched ionic channels [23,108]. However, its mechanical properties are to be fine-tuned for its application as PEM in DMFCs as it swells significantly at a high sulfonation level [109]. Researchers have explored various routes to overcome this issue by taking properties of GO into an advantage and forming the composite membranes with SPEEK. Some of the important reports are discussed below.

GO was functionalized with SDBS wherein it is adsorbed on the surface of GO through π-π and hydrophobic interactions. Incorporation of SDBS-GO in SPEEK reduces the dimension of the ionic cluster from 1.96 to 1.76 nm, thus mitigating the methanol permeability [54]. The reduction in ionic cluster is mainly associated with increased interactions of benzene rings in SDBS and hydrophobic backbone and sulfonic acid groups of SDBS and hydrophilic clusters of SPEEK. In addition, proton conductivity is enhanced by the incorporation of GO/SDBS-GO. Unlike Nafion^®^-GO composite membranes, wherein incorporation of GO reduces the proton conductivity, enhanced proton conductivity is observed for SPEEK-GO composite membranes, as the incorporation of GO improves the inter-connectivity between the ionic channels. This distinct behaviour is also explained based on the microstructure of Nafion^®^ and SPEEK. Compared to Nafion^®^, SPEEK ionic channels are narrow and more branched and not well connected (dead-end channels) [108]. It is worth-noting that incorporation of GO (5 wt.%) in SPEEK enhances the proton conductivity from 39.5 mS cm^−1^ to 53.4 mS cm^−1^, but incorporation of SDBS-GO (the same 5 wt.%) further enhances proton conductivity to 79.4 mS cm^−1^. This effect is ascribed to the additional sulfonic acid groups present in SDBS which enhance not only the connectivity between the ionic channels but also create additional channels for proton hopping.

Another method of sulfonating GO is by using propane sultone. Unlike in SDBS-GO, in this case, there is a chemical bond between GO and sulfonic acid [55]. By using this sulfonated GO as an additive in SPEEK there are certain differences in properties of the composite membranes in comparison with SPEEK-SDBS-GO attributed to the difference in the functionalization method. SPEEK-SGO composite membrane shows a maximum proton conductivity of 85 mS cm^−1^. However, beyond the optimum loading, proton conductivity decreases due to the blockage of proton-conducting channels by forming the agglomerates of GO/SGO. The best selectivity was seen for SPEEK-SGO (7 wt.%), wherein the highest proton conductivity and lower methanol permeability were observed.

In addition, GO was also sulfonated using benzene sulfonic acid. This is a two-step reaction, in the first step, sulfanilic acid is treated with hydrochloric acid and sodium nitrate to form 4-benzenediazonim salt. In the second step, the formed precursor is then treated with GO to form sulfonic acid functionalized GO through new C-C bond between sp^2^ carbon of GO and sp^2^ carbon of benzene sulfonic acid. When this S-GO is used as additive in SPEEK, as reported in the literature, SGO improves the homogeneous dispersion of GO in SPEEK due to enhanced compatibility between sulfonated groups of GO and SPEEK. Further, the improvement in mechanical properties is observed as a result of strong interfacial interactions between graphitic planes of S-GO and SPEEK matrix. However, GO being a sheet-like structure with large aspect ratio its orientation in the membrane matrix is critical in influencing the properties of composite membranes especially in terms of proton transport. When the basal planes of GO are oriented parallel to the plane of the membrane may act as a barrier for proton transport in the membrane. The introduction of holes in GO sheet is a promising approach that can construct the new pathway for proton transport, Jiang et al. used different types of GOs *viz*; GO, S-GO holy-GO (H-GO) sulfonated holy GO (SH-GO) in SPEEK metrics and observed the properties [57]. H-GO and SH-GO shows superior proton conductivity (71 mS cm^−1^, 136 mS cm^−1^ respectively) than GO and S-GO (56 mS cm^−1^, 92 mS cm^−1^, respectively) proving that H-GO sheets are beneficial towards proton transport on the other hand methanol diffusion also improved for H-GOs but still the overall selectivity is higher for H-GO composites.

Our recent study explored the use of functionalized GO as a potential additive in SPEEK membranes [110]. Unlike previous reports, GO was functionalized with amino acid wherein proton transport occurs via amine groups and carboxylic acid groups. On the other hand, amino acid forms electrostatic interactions with water molecules and held between –NH_2_^+^– and –COO^−^ groups. These bridged water molecules form a hydration layer in the composite membrane acting as a vehicle for proton transport [111]. Thus, a small quantity of amino acid functionalized GO (1 wt.% in relation to SPEEK) brings significant improvement in proton conductivity. Coming to the methanol permeability, there is a 40% reduction in methanol permeability of this composite membrane compared to pristine SPEEK, this should be attributed to the barrier effect of GO.

#### 2.2.3. Sulfonated Poly Imide-GO Composites

Sulfonated polyimide (SPI) is also one of the most explored PEMs for DMFCs due to its hydrolytic stability and action as a potential methanol barrier and are resistant to nucleophilic attack due to the presence of naphthalenic moieties [61,112,113,114]. The distinct physico-chemical properties, such as structural anisotropy, the partial separation between their hydrophobic and hydrophilic domains, the constant hydration level (λ) over a wide range of ion content, and a multi-scale foliated structure packed along the membrane thickness are beneficial, resulting in performance comparable to Nafion^®^. However, imide rings are sensitive to hydrolysis under hydrated conditions resulting in poor water/methanol stability. To address the stability concern and proton conductivity, functionalized GOs such as sulfonated propyl silane GO (SPSGO) [59] or polystyrene sulfonic acid GO (PSSGO) [60] have been explored as potential additives to SPI. Incorporation of SPSGO in SPI improves the bound water content due to the formation of additional ionic clusters from SPSGO (the authors report λ = 15.1 for SPI with 8 wt.% SPSGO, and λ = 9.82 for pure SPI. Due to the improved water retention capacity of SPI-SPSGO membranes, it is possible to operate at elevated temperatures. A DMFC test of SPI-SPSGO membranes at 130 °C yielded a peak power density of 100 mW cm^−2^, confirming the high temperature operation capability. The stability of the composite membrane is evaluated in a single-cell by measuring the current density of the cell operating at 0.6 V to study the degradation of membrane due to hydrolysis, radical attack. However, the test has been conducted only for 70 h and the observed marginal performance drop, the long-term stability (life-time) of this membrane should be as adequately studied as it is for PEMs in DMFCs. In addition to this, Chi-Yung Tseng et al. prepared a composite membrane of SPI and PSSGO (PSSA grafted on GO) with an improved sulfonic acid content of the membrane [60]. This improves the ionic conductivity and the GO acts as a barrier for methanol transport through the PEM, leading to better selectivity for SPI-PSS-GO membranes than commercial Nafion-117 (Figure 12).

#### 2.2.4. Sulfonated Poly Sulfones-GO Composites

Nanocomposite membranes of sulfonated polysulfone are explored with varies types of GO to resolve the issues related to methanol permeability, mechanical stability and swelling at high degrees of sulfonation [62,63,64,65]. The loading of GO can be optimized on the basis of functional groups present on the surface of GO. For example, incorporation of more than 1 wt.% pure GO in sulfonated polyether sulfone (SPES) shows a blocking effect on proton conductivity, whereas sulfonated GO and ionic liquid functionalized GO composites can be prepared up to 5 wt.% of loading. This is due to the formation of interfacial interactions between the nano-filler and SPES. Further, the dispersion of GO is more likely to be a physical blending whereas, in the case of functionalized GO, there are inherent interfacial interactions between functional groups and sulfonic acid groups of SPES. Figure 13 illustrates these interactions between SPES and ionic liquid functionalized GO.

These types of electrostatic interactions are also observed in SPES-SGO composites wherein hydrogen bonding networks are formed between sulfonic acid groups of SGO and SPES [64]. Proton conductivity is also greatly influenced by the nature of functional groups, as the mechanism of ionic conduction in each of them is very different. Conversely, the mechanism of methanol permeability is almost similar as it is mainly dominated by the blocking effect of GO. It is observed that the composite membranes of SPES show a raise in selectivity up to 33% with simple GO, 46% with sulfonated GO and 42% with ionic liquid functionalized GO in comparison to their pristine membranes. The difference is due to the functional groups attached on the surface of GO and their interaction with the polymer matrix. Sulfonic acid groups generally aggregated to form clusters due to electrostatic and or acid–base interactions between the nano-filler and SPES polymer. With the addition of GO/functionalized GO there is an obvious increment in d-spacing resulting in the enlargement of ionic clusters favouring proton transport [64].

#### 2.2.5. PVA-GO Composite PEMs

As described above, PVA is widely studied for DMFC application due to its ease of processing and water-methanol separation characteristics. However, its poor mechanical stability and high swelling hamper its use as an electrolyte in DMFCs [90,91,92,115,116,117]. In this context, GO is identified as potential nano-filler to improve the properties of PVA membranes, GO-incorporated PVA films show lower water uptake than pure PVA membrane due to the interaction of carboxylic groups of GO with the hydroxyl groups of PVA. In addition, methanol permeability also followed the similar trend wherein the methanol permeability of 1.5 wt.% GO incorporated PVA is 64% lower than for a pure PVA membrane [72]. Another report on PVA-GO composite membrane describes the preparation of iron oxide deposited sulfonated graphene oxide (Fe_3_O_4_-SGO) incorporated in PVA [73]. In this study, the composite membranes were prepared by applying the external magnetic field to the PVA-Fe_3_O_4_-SGO composite solution while forming the membrane. The applied magnetic field enables the orientation of SGO nano-sheets through the plane of the membranes (Figure 14). This perpendicular orientation of SGO sheets in the membrane provide facile proton transport by forming wider ionic channels but, On the other hand, it also increases the methanol permeability in comparison with the randomly oriented SGO composites membranes. However, the overall selectivity is higher for the former and shows a DMFC power output 23% higher than the latter.

#### 2.2.6. GO Free-Standing Membranes

Apart from composite membranes of GO, few reports are available on the fabrication of GO paper and its application as PEM in DMFCs [74,75]. The underlying design principle benefits from the intrinsic low electronic conductivity of GO, which can be functionalized with acidic groups on the surface to enable ion conduction while maintaining the methanol impermeable characteristics [118,119,120]. Generally, these GO membranes are prepared by vacuum filtration of GO colloidal dispersion (Figure 15). The performance of these type of membranes mainly depends on the flake size of GO. Abhilash Paneri et al. observed a linear relationship between flake size and methanol permeability whereas proton conductivity is not much altered with varying flake size. The stability of GO membranes is higher than Nafion^®^ even at methanol concentrations as high as 10 M [75,121,122,123]. Zhongqing Jiang et al. reported a series of GO papers namely GO, HGO, SDBS-GO and SDBS-HGO for air-breathing DMFCs [74]. As explained above, GO paper shows good methanol resistant behavior with reasonable proton conductivity. However, the electrochemical performance of GO paper is further improved by altering its microstructure by fabricating HGO and functionalized GO. These HGO, SDBS-GO and SDBS-HGO membranes show improved DMFC performance compared to GO paper. One highlights the high performance of thee SDBS-HGO-based composite membrane, which shows 23% higher values than Nafion-112. Such impressive enhancement is further verified by the stability test under constant load of 50 mA cm^−2^ where SDBS-HGO shows higher stability than Nafion-112 and, thus, better lifetime can be expected for these types of GO papers.

### 2.3. PEMs with Other Carbon Nanomaterials

Unlike CNTs and GO, fullerene is not well explored as an additive for PEMs due to its bulky nature and lack of functionalization routes to disperse it in the polymer matrix. However, in spite of all these, few of the research groups have studied the effect of fullerene as an additive for PEMs by considering its advantages like high electron affinity, high volumetric density of the surface functional group and radical scavenging properties along with its high thermo-mechanical properties [124]. Saga et al. prepared nanocomposite membrane of sulfonated polystyrene with fullerene as additive and studied its properties in terms of mechanical and chemical stability, methanol permeability and proton conductivity. It is observed that fullerene acts as an effective barrier for methanol transport and improves chemical stability due to its radical scavenging property, wherein hydroxyl (•OH) and hydroperoxyl (•OOH) radicals are trapped by fullerene produced during DMFC operation causing membrane degradation. On the other hand, the mechanical stability and proton conductivity of the membrane are not improved due to the poor compatibility and lack of ion-conducting groups in fullerene [125]. Recently we explored a new strategy to functionalize fullerene with 4-benzene diazonium sulfonic acid (Figure 16a) to use as a nano-additive in PEMs. This sulfonation route of fullerene improves its dispersion and compatibility in polymer matrices and forms proton-conducting sites. Nanocomposite membranes prepared by dispersing S-fullerene in SPEEK and Nafion^®^ polymersdisplay improved stability in methanol environment (Figure 16b), which is attributed to its better methanol-tolerant characteristics [76,77].

The optimized SPEEK-Sfu (0.5 wt.%) membrane shows a peak power density of 103 mW cm^−2^, which is 44% higher than pristine SPEEK shown in Figure 17a. In addition to its better DMFC power output, the optimized S-fullerene composite membranes also show better oxidative stability than pristine SPEEK due to the radical scavenging property of fullerene as shown in Figure 17b [76]. The effect of functionalized fullerene (FF) as an additive in Nafion^®^ membrane in DMFC is also studied wherein controlling methanol permeability is one of the major challenges due to the larger ionic cluster of Nafion^®^ [77].

The Nafion^®^-Sfu composite membrane shows methanol permeability reduced up to 30% with 2 M methanol, 23% with 3 M methanol and 38% with 5 M methanol in comparison with pristine Nafion^®^ membrane. This is attributed to the occupation of the ionic cluster by S-fullerene thereby increasing the tortuosity of the pathways for methanol transport. The reduction in methanol permeability is replicated in the power density curves shown in Figure 18, wherein Nafion^®^-S-fullerene shows higher performance at all concentrations of methanol in comparison with pristine Nafion^®^.

In addition to the above materials, other carbon nanomaterials, like carbon black (CB) and graphite nanofibres (GNF), are also studied as additives in PEMs. For example, Y.S. Ye et al. [126] studied the effect of ionic liquid functionalized carbon black as an additive in sulfonated polyimide matrix. The composite membranes are characterized for through-plane and in-plane ionic conductivity. It is found that addition of a small amount of functionalized CB (0.2 wt.%) enhances the conductivity significantly due to the acid-base interactions between imidazole groups on the surface of CB and sulfonic acid groups of the polymer. In-plane activation energy was reduced from 44 kJ mol^−1^ to 38 kJ mol^−1^ while the through-plane activation energy was reduced from 46 kJ mol^−1^ to 31 kJ mol^−1^ indicating that addition of functionalized CB to polymer matrix in an appropriate quantity can greatly improve the transport properties. Xupo Liu et al. explored the synthesis and sulfonation of carbon nanofibres (CNFs) and its use as additives in SPEEK as PEM in DMFCs [78]. These membranes show higher proton conductivity compared to pristine membrane due to their morphological difference. Figure 19a,b shows AFM images for the membranes wherein composite membrane show good phase separation between the hydrophobic backbone and self-aggregated hydrophilic regions resulting in facile proton transport. In addition to this, the composite membrane show reduced methanol permeability as low as 5.02 × 10^−7^ cm^2^ s^−1^ leading to improved electrochemical selectivity.

Our recent reports explored the effect of sulfonated graphite nanofibres (SGNF) as an additive in sulfonated poly(phthalalizone ether ketone) (SPPEK) matrix [77]. The composite membrane show superior properties in terms of proton conductivity and mechanical stability. Methanol permeability for the membranes is greatly reduced due to the incorporation of SGNF. The optimized composite membrane (SPPEK-SGNF (0.5 wt.%)) show a peak power density of 115 mW cm^−2^ in comparison with pristine SPPEK, which shows 38 mW cm^−2^. The methanol crossover current density through the MEAs comprising pristine SPPEK and SPPEK-SGNF was measured by linear sweep voltammetry (LSV) after 50 h OCV operation. It is found that pristine SPPEK shows a two-fold increase in crossover current density with time, whereas the composite membrane shows a marginal increment in the crossover current density suggesting its methanol-blocking characteristic and greater stability of the composite membranes under DMFC operating conditions.

## 3. Conclusions

Carbon nanomaterials have created a significant impact on the development of polymer electrolytes for DMFCs. The performance of composite membrane electrolyte is influenced by the choice of carbon nanomaterial, i.e., graphene oxide (two-dimensional), CNTs, CNF/GNF (uni-dimensional) and fullerene. Structural orientation and composition of these carbon materials in the membrane matrix has a great impact on the properties of membranes. Surface functionalization for these materials impart the hydrophilic nature and fine-tunes its conductivity to provide better interfacial interactions with the polymer matrices. These functional groups enable the facile ion transport improving the ionic conductivity of the composite membranes, hence, the carbon structure and functionalization methods should be carefully chosen for the optimum cell performance. For Nafion^®^ membranes GO is the most suitable nano-filler wherein mitigation of methanol permeability has prime importance, the sheet-like structure is more effective than the tubular structure in arresting the fuel permeability, hence, GO is preferentially chosen to prepare Nafion^®^ composite membranes, whereas, for hydrocarbon polymers, functionalized CNTs are appropriate fillers wherein improvement in ionic conductivity and mechanical strength are necessary. Hydrocarbon PEMs have significantly lower methanol permeability than Nafion^®^, hence their proton conductivity and mechanical properties should be raised to realize their commercial use. CNTs are appropriate fillers to improve the mechanical stability but, again, the proton conductivity is another limitation which can be eliminated by functionalizing the nano-filler with proton-conducting groups.

## 4. Future Prospects

Carbon nanomaterials have proved to be promising reinforcing materials in PEMs for fuel cell technology in which carbon composites have recently emerged as promising materials. By the proper tuning of their properties, and adequate use of carbon fillers in PEMs, they can provide solutions for most of the current limitations in PEMs and help in advancing the fuel cell technology to be available for day-to-day utilization. On the other hand, one of the key issues is that the lifetime of these carbon-nanocomposites is not adequately addressed in the literature so far, and stability tests are hardly carried out up to a few hundred hours. However, for commercial DMFC the expected lifetime is much higher, hence future studies need to clarify this point to progress carbon-polymer nanocomposite PEMs for DMFCs. Nevertheless, carbon nanofillers are promising materials in many other technologies, such as water filtration, desalination and flow batteries. However, large-scale, cost-effective production is the key challenge, in addition to their dispersion, hence adequate investigations should be carried out in terms of low-cost production and improved dispersion to extend their uses to numerous technologies.

## Figures and Tables

**Figure 1 nanomaterials-09-01292-f001:**
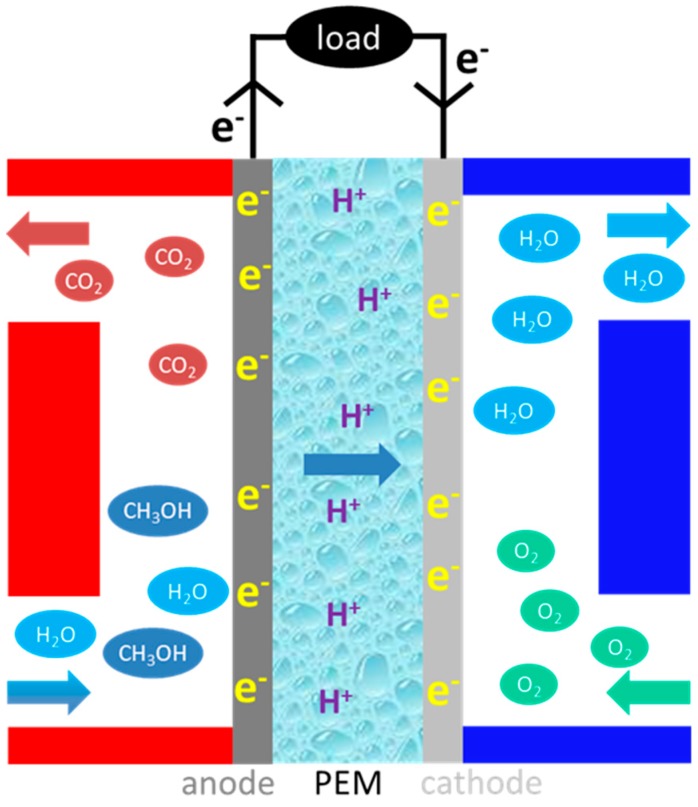
Schematic diagram of a DMFC comprising the anode (Pt-Ru/C) and the cathode (Pt/C), separated by a polymeric proton exchange membrane (PEM).

**Figure 2 nanomaterials-09-01292-f002:**
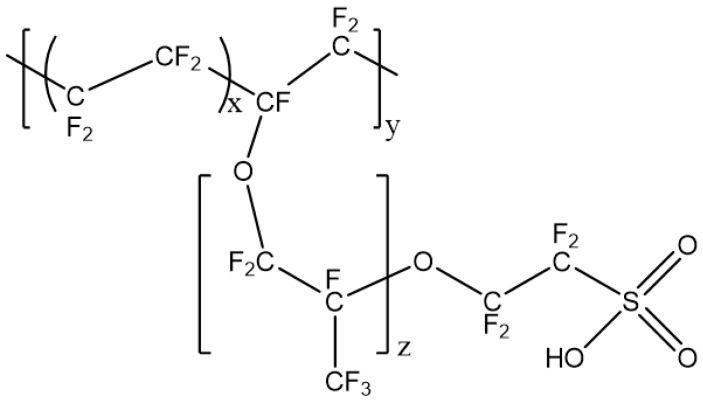
Chemical structure of Nafion^®^.

**Figure 3 nanomaterials-09-01292-f003:**
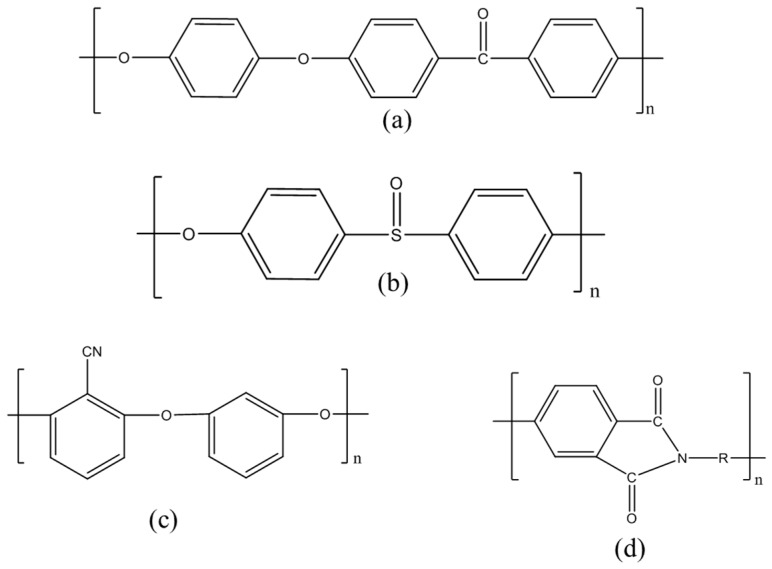
Chemical structures of (**a**) PEEK, (**b**) PES, (**c**) PEN and (**d**) cyclic PI.

**Figure 4 nanomaterials-09-01292-f004:**
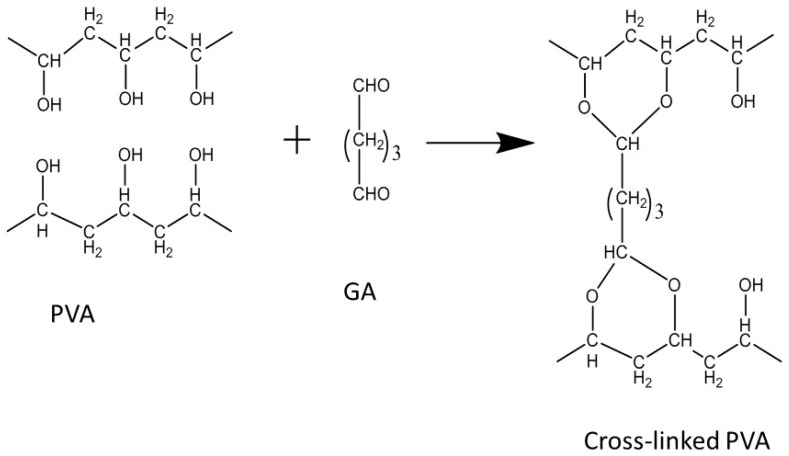
Crosslinking of PVA using GA.

**Figure 5 nanomaterials-09-01292-f005:**
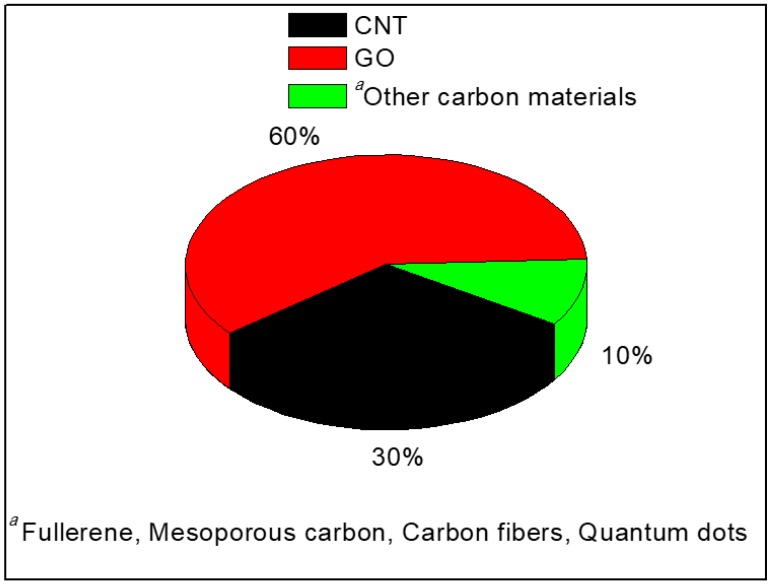
Percentage of GO, CNT and other carbon-based membrane electrolytes reported during last ten years. The data is generated by considering the references listed in Table 1.

**Figure 6 nanomaterials-09-01292-f006:**
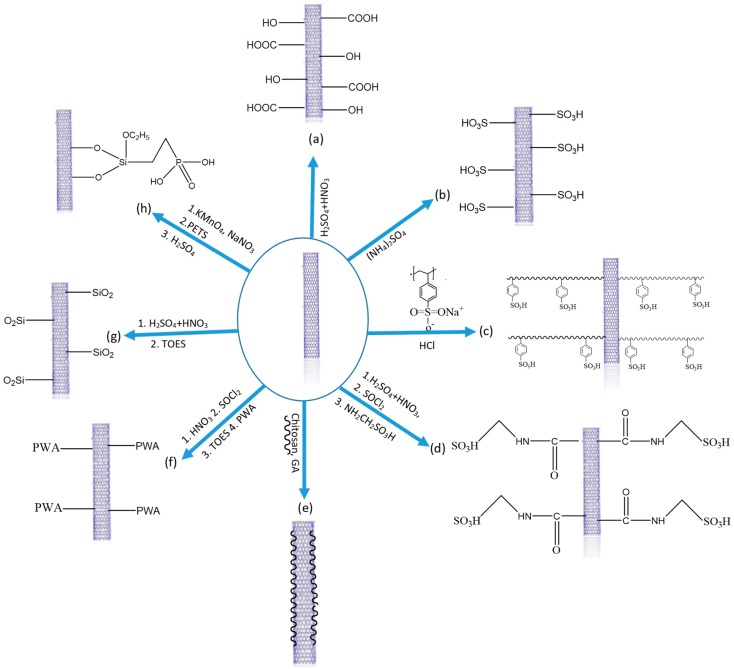
Different functionalization approaches for CNTs using (**a**) carboxylic acid, (**b**) sulfonic acid (**c**) PSSA, (**d**) aminomethanesulfonic acid, (**e**) chitosan, (**f**) phosphotungstic acid (**g**) silica and (**h**) phosphonic acid groups.

**Figure 7 nanomaterials-09-01292-f007:**
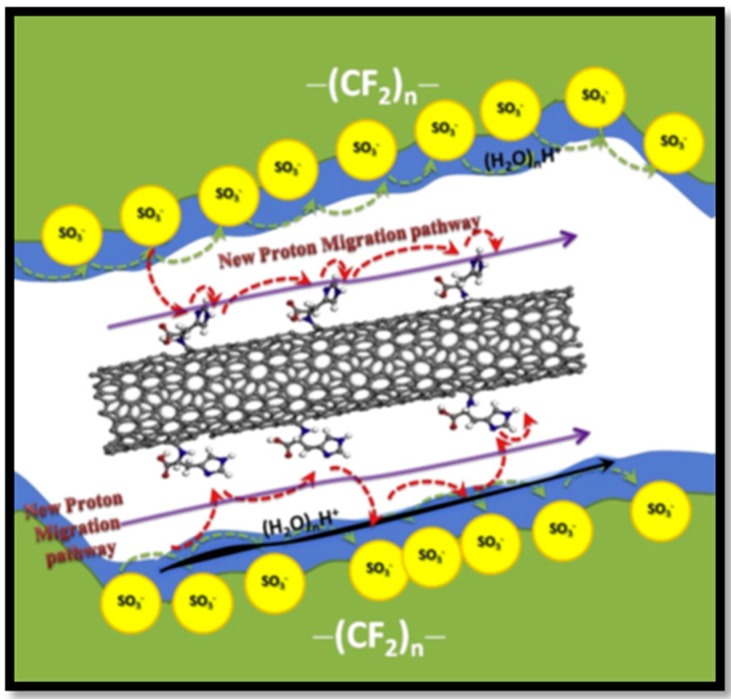
Proton conduction mechanism in Nafion^®^-ImCNTs composite membrane. Reproduced from [29] with permission from Elsevier, 2013.

**Figure 8 nanomaterials-09-01292-f008:**
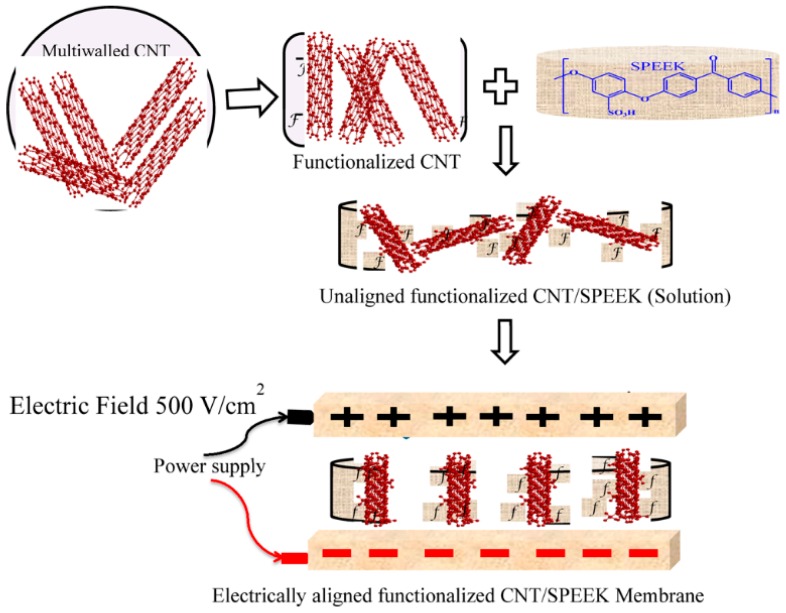
Schematic representation of electrically aligned CNTs in SPEEK matrix. Reproduced from [41], with permission from American Chemical Society, 2015.

**Figure 9 nanomaterials-09-01292-f009:**
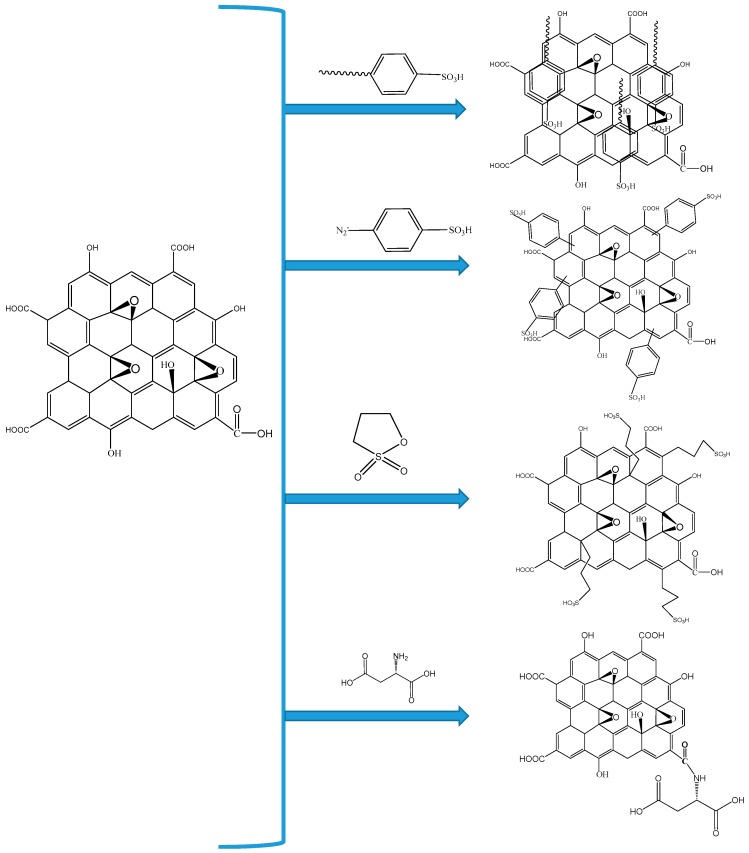
Different functionalization methods for GO.

**Figure 10 nanomaterials-09-01292-f010:**
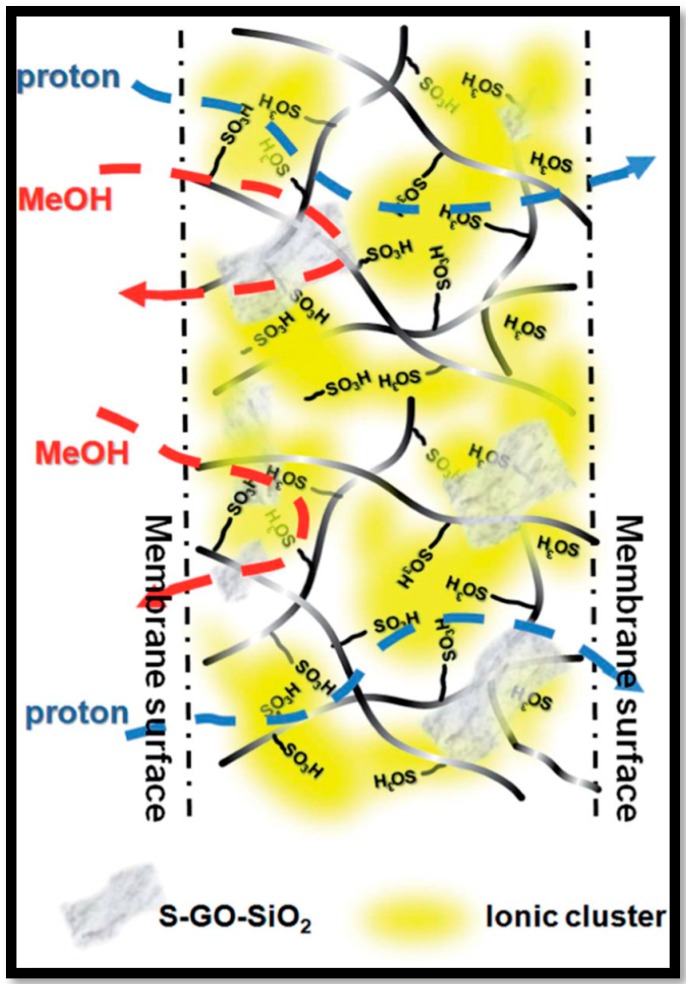
Schematic illustration of proton and methanol transport in S-GO–SiO_2_/Nafion^®^ composite membrane. Reproduced from [48], with permission from Royal Society of Chemistry, 2014.

**Figure 11 nanomaterials-09-01292-f011:**
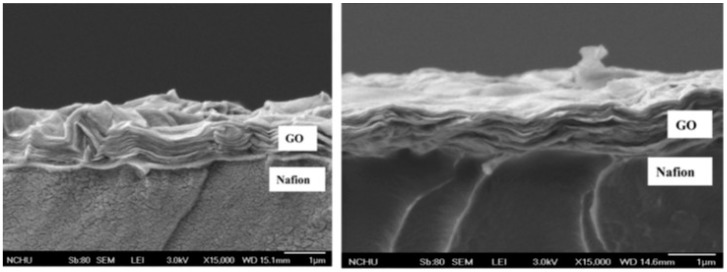
SEM images of the GO-laminated Nafion^®^ membrane at different magnitudes. Reproduced from [72], with permission from Elsevier, 2013.

**Figure 12 nanomaterials-09-01292-f012:**
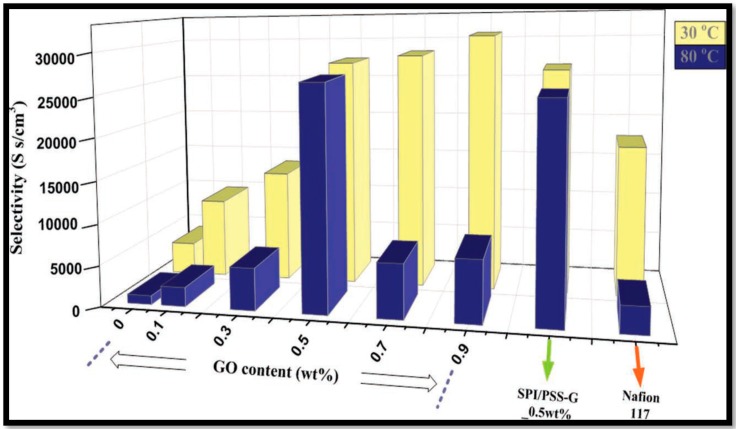
Selectivity of SPI and SPI-PSS-GO composite membranes compared with Nafion-117 membrane. Reproduced from [60], with permission from John Wiley and Sons, 2011.

**Figure 13 nanomaterials-09-01292-f013:**
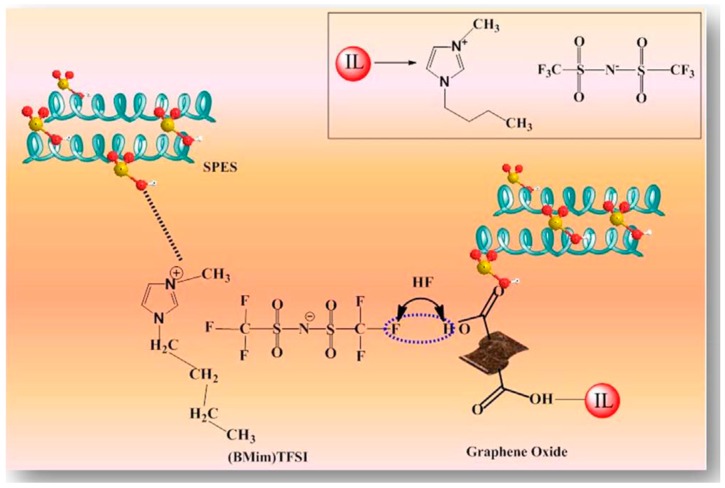
Schematic representation of interaction between IL-GO and SPES matrix. Reproduced from [63], with permission from Elsevier, 2018.

**Figure 14 nanomaterials-09-01292-f014:**
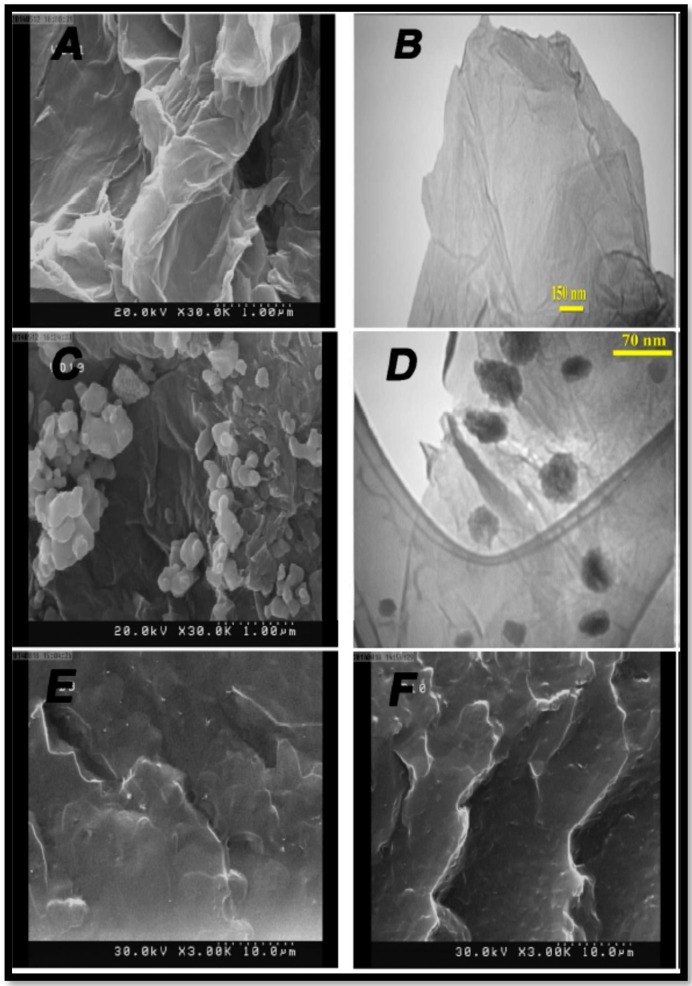
SEM and TEM images of (**A**,**B**) SGO and (**C**,**D**) SGO/Fe_3_O_4_ nanosheets and cross-sectional SEM images of (**E**) PVA-GLA-SGO-Fe_3_O_4_ (**F**) PVA-GLA-SGO membranes. Reproduced from [93], with permission from American Chemical Society, 2015.

**Figure 15 nanomaterials-09-01292-f015:**
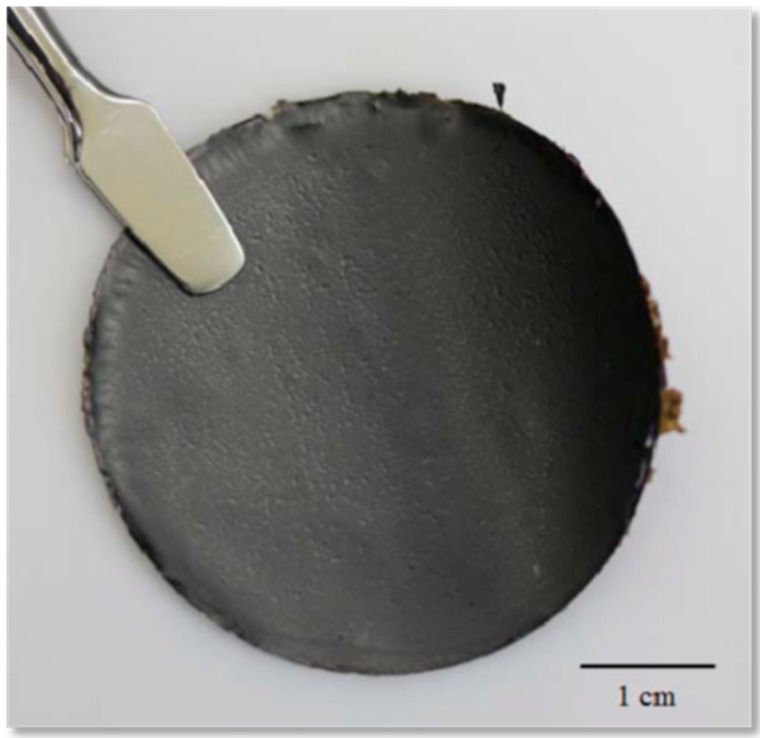
Photograph of a 12-μm-thick standalone GO laminate prepared by vacuum filtration of GO nanoplatelets suspension in DI water. Reproduced from [75], with permission from Elsevier, 2014.

**Figure 16 nanomaterials-09-01292-f016:**
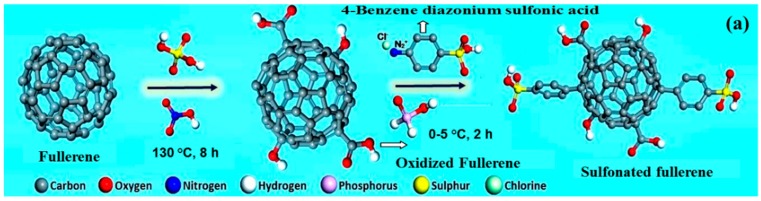
(**a**) Schematic representation of sulfonation procedure for fullerene and (**b**) tensile strength of SPEEK-S-fullerene membranes after equilibrating in different concentration of methanol (0–8 M). Rreproduced from [76], with permission from Elsevier, 2016.

**Figure 17 nanomaterials-09-01292-f017:**
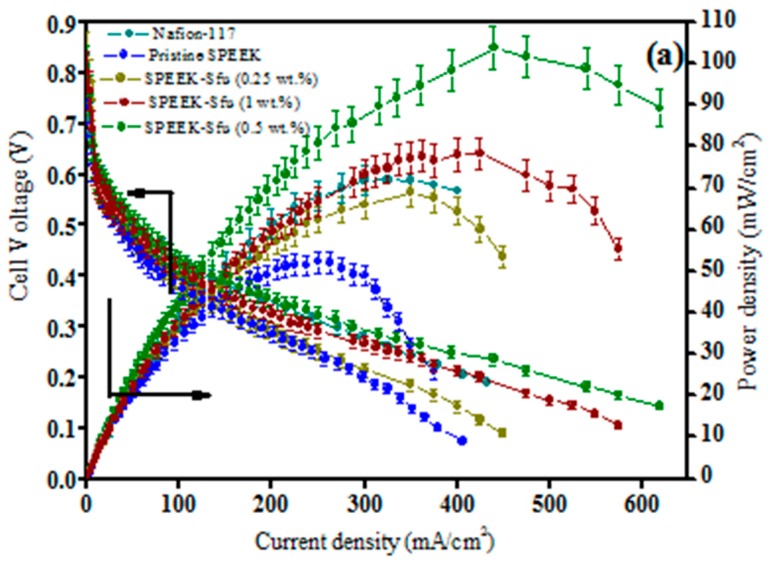
(**a**) DMFC performance representing steady-state cell polarization for Nafion-117, pristine SPEEK and SPEEK-S-fullerene composite membranes at 60 °C and (**b**) oxidative stability for membranes. Reproduced from [76], with permission from Elsevier, 2016.

**Figure 18 nanomaterials-09-01292-f018:**
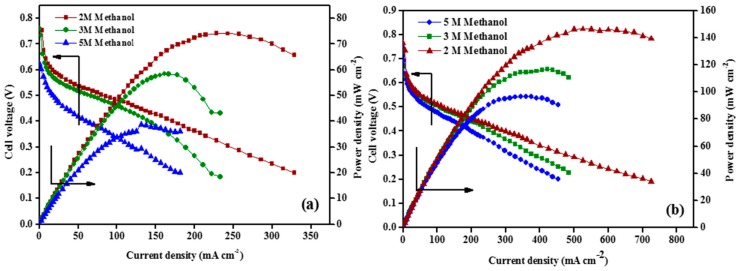
DMFC polarization for (**a**) recast Nafion^®^ and (**b**) Nafion^®^-FF (1 wt.%) at different methanol concentrations at 60 °C (anode: Pt-Ru/C, 2 mg cm^−2^ and cathode: Pt/C, 2 mg cm^−2^. Anode fuel: 2 M methanol 2 mL min^−1^, cathode: oxygen 300 mL min^−1^). Reproduced from [77], with permission from Elsevier, 2016.

**Figure 19 nanomaterials-09-01292-f019:**
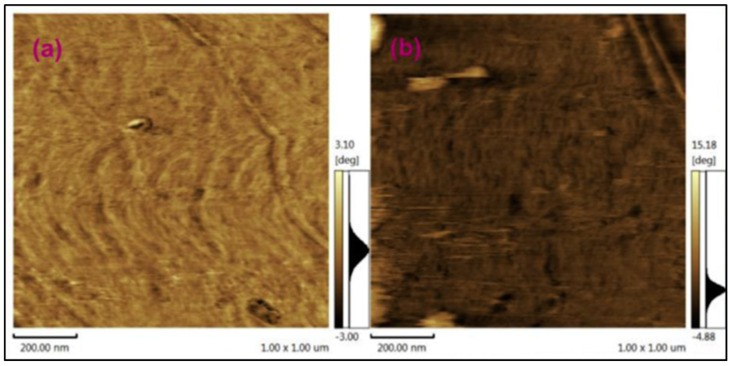
AFM phase images for (**a**) pristine SPEEK, and (**b**) SPEEK-SCNF (1 wt.%) membranes. Reproduced with permission from [78], with permission from Elsevier, 2017.

**Table 1 nanomaterials-09-01292-t001:** Summary of properties of polymer electrolytes reported in the literature using carbon nanomaterials as additives.

Polymer Matrix	Carbon Material	Additive Loading wt.%	Proton Conductivity (mS cm^−1^)	Methanol Permeability (×10^−7^ cm^2^ s^−1^)	Temp. (°C)	Ref.
Nafion^®^	Im-CNT	0.5	150	10.0	80	[29]
Nafion^®^	COOH-MWCNT	2	100	11.3		[30]
Nafion^®^	Chitosan-CNTs	0.5	104	2.03	25	[31]
Nafion^®^	PWA-SiO_2_-CNT	1	87	2.63	25	[32]
Nafion^®^	Fe_3_O_4_-CNTs	0.1	85	5.4	60	[33]
Nafion^®^	CeO2−ACNTs	2	96	NR	60	[34]
SPEI	S-MWCNT	5	3.98	11.7	80	[35]
SPAS	SO_3_CNT/PtRu/CNT	1	106	23.6		[36]
PVA	s-MWNTs	20	75	0.03	60	[37]
PVA	S-MWCNT/F-MMT	1	6	20	30	[38]
PVA	S-MWCNT	1	4	41	30	[38]
PVA	CNT-PDDA-HPW	2	9.4	4.02	30	[39]
SPESEKK	sCNTs	1.5	4.3	0.96	30	[24]
SPEEK	PSSA-CNTs	0.5	101	2.17	60	[40]
SPEEK	fCNTs	0.5	43.1	1.68	30	[41]
SPEEK	POH-CNTs	2	160	3.7	60	[42]
SPEEK	SiO_2_-CNTs	1.5	77.8	0.72	RT	[43]
SDBC	S-CNTs	1.5	141.7	NR	90	[44]
Nafion^®^	GO	1.5	23.5	9.1	35	[45]
Nafion^®^	S-GO	0.5	100	19.9	60	[46]
Nafion^®^	GO	0.5	40	7.92	30	[47]
Nafion^®^	GO-silica	0.8	48.1	0.16	50	[48]
Nafion^®^	S-GO	NR	89.6	8.4	30	[49]
Nafion^®^	Graphene	NR	40	4.4	25	[50]
Nafion^®^	GO	NR	15	0.67	30	[51]
Nafion^®^	PDDA-GO	NR	25	13	25	[52]
SPEEK	S-GO	8	162.6	13.6	65	[53]
SPEEK	SDBS-GO	8	162.6	9.5	65	[54]
SPEEK	S-GO	5	8.41	2.6	80	[55]
SPEEK	Histidine-GO	4	69.4	1.35	25	[56]
SPEEK	SH-GO	5	90.5	0.3	25	[57]
SF-SPEEK	GO	5	111.90	NR	90	[58]
SPI	SPS-GO	8	96.2	2.0	30	[59]
SPI	PSS-GO	0.5	86	4.31	60	[60]
SPI	SI-GO	10	113.8	10.52	30	[61]
SPES	S-GO	5	58	1.5	30	[62]
SPES	IL-GO	5	72.7	0.53	RT	[63]
SPES	S-GO	15	78.2	3.82	25	[64]
SPES	GO	1	4.3	0.49	RT	[65]
SPE	S-GO	0.75	390	4.89	80	[66]
SPS	S-GO	3	4.27	3.48	RT	[67]
SPEN	N-GO	1	104	1.74	20	[68]
SPEN	S-N-GO	NR	64	1.43	20	[69]
SPAEN	CNT-GO	NR	119.7	2.0	20	[70]
SPVdF-co-HFP	S-GO	0.7	5.5	1.8	30	[71]
PVA	GO	1.5	13.5	2.0	35	[72]
PVA	Fe_3_O_4_/S-GO	5	64	0.45	30	[73]
	GO paper		54.2	2.4	65	[74]
	Holey GO paper		68.4	145.5	65	[74]
	SDBS-GO paper		68.5	4.4	65	[74]
	Holey SDBS Holey GO paper		91.8	35.4	65	[74]
	GO paper		4.9	0.16	30	[75]
SPEEK	S-fullerene	0.5	96	2.4	60	[76]
Nafion^®^	S-fullerene	1	97	8.5	60	[77]
SPEEK	SCNF	1	128	5.02	60	[78]
SPPEK	SGNF	0.5	85	4.5	60	[79]
Nafion^®^	MC	1	75	9.8	30	[80]
Nafion^®^	NCD	0.5	21	0.12	40	[81]

NR: Not reported.

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
