# Peer review of "Carbon Nanocomposite Membrane Electrolytes for Direct Methanol Fuel Cells—A Concise Review"

_nanomaterials, 2019, doi:10.3390/nano9091292_

Round 1
Reviewer 1 Report
The manuscript is well organized and in its present form reviews in a proper way the related literature. The experimental data are reviewed and presented in a clear and lucid way and the conclusions are rationalized.
Author Response
Comments Reviewer 1:
The manuscript is well organized and in its present form reviews in a proper way the related literature. The experimental data are reviewed and presented in a clear and lucid way and the conclusions are rationalized.
Authors response: We thank the reviewer for spending his/her valuable time on reviewing the manuscript and his positive remarks on the manuscript.

Reviewer 2 Report
The paper summarize the recent development of carbon based membrane for direct methanol fuel cells application. A wide range of materials is presented. This work is very well written. I would suggest for publication, with small improvements on the quality of the figure 3 and figure 4. Please check the English language before publishing.
Author Response
Comments Reviewer 2:
The paper summarize the recent development of carbon-based membrane for direct methanol fuel cells application. A wide range of materials is presented. This work is very well written. I would suggest for publication, with small improvements on the quality of the figure 3 and figure 4. Please check the English language before publishing.
Authors response: We thank the reviewer for spending his/her valuable time on reviewing the manuscript and his positive remarks on the manuscript. As per the reviewer suggestion quality of Figure 3 and Figure 4 are improved. We also checked the English language and corrected in the revised manuscript.
Reviewer 3 Report
Dear authors,
your review on DMFCs is fine, however in my opinion it deserves minor corrections in order to increase its impact for the fuel cell community in general.
It would take the introduction to make a presentation a little more specific DMFC and their use with performance wait for this type of stack,then make a paragraph the materials of the state of the art including putting the axon on the membrane, highlighting the interest of Nafion and other derived compounds presented in your introduction, and then initialing on the bolts to overcome for have a use and finish this initials by the interest of carbon-based nanocomposites indeed, according to the title and the content of your review is the central point of your publication but it is not clearly highlighted the interest and the stake of the use of such a compounds in DMFCs;
Finally, I find that your descritpion of each material is well made as well as the examples of use in pile configuration but it is list because of the lack of clear exposure of the objectives, locks to lift for the membranes. You never talk about the life of its compounds, can you specify this point in your review?
Best regards
reviewer
Author Response
Comments Reviewer 3:
Comment 1: Dear authors, your review on DMFCs is fine, however in my opinion it deserves minor corrections in order to increase its impact for the fuel cell community in general.
Authors response: We thank the reviewer for spending his/her valuable time on reviewing the manuscript and his positive remarks on the manuscript.
Comment 2: It would take the introduction to make a presentation a little more specific DMFC and their use with performance wait for this type of stack.
Authors response: As per reviewer suggestion introduction is revised and added some more details about DMFC.
The transport sector contributes to most of the atmospheric pollution and consumes of the major portion of the energy generated world-wide. On the other hand portable electronic market such as smartphones, laptops in growing rapidly and state of art Li-batteries are lagging in some aspects including safety. A portable DMFC system is expected provide solutions to these mobile electronics. Hence the development of an ideal DMFC system is prime importance in the current fuel cell research.
This is given in section 1, page 1, and paragraph 1 of the revised manuscript and highlighted in yellow.
Comment 3: make a paragraph the materials of the state of the art including putting the axon on the membrane.
Authors response: All the state of art materials for anode, PEM and cathode are mentioned.
DMFC comprises of many components namely membrane electrode assembly (MEA), flow channels, endplates, current collector and among all the components of DMFC, the membrane electrode assembly MEA is the key component comprises a polymer electrolyte membrane sandwiched between an anode and cathode. The state of art PEM is Nafion, a perfluorosulfonic acid (PFSA) membrane and anode material is Pt-Ru bimetallic catalyst supported on carbon while the state of art cathode is Pt supported on carbon.
This is given in section 1, page 1, paragraph 1 of the revised manuscript and highlighted in yellow.
Comment 4: initialing on the bolts to overcome for have a use and finish this initials by the interest of carbon-based nanocomposites indeed, according to the title and the content of your review is the central point of your publication but it is not clearly highlighted the interest and the stake of the use of such a compounds in DMFCs;
Finally, I find that your descritpion of each material is well made as well as the examples of use in pile configuration but it is list because of the lack of clear exposure of the objectives, locks to lift for the membranes
Authors response: Explanation about Nafion and its drawbacks are highlighted
Nafion® membrane is used as a current state-of-the art PEM for DMFCs for its remarkable mechanical and chemical stability and high proton conductivity. Nafion® consist a strong hydrophobic fluorinated backbone and hydrophilic pendant sulfonic acid chain.
The backbone offers remarkable mechanical strength along with chemical stability, and the sulfonic acid groups improve the water retention capacity and are responsible for its superior protonic conductivity [5]. However, the aqueous domains formed in the vicinity of these ionic clusters also lead to high methanol permeation from the anode to the cathode, where it is oxidized creating a mixed potential that reduces the overall efficiency of the cell [6, 7]. Hence, research efforts have been made to overcome the above issues by two main different approaches viz., modification of PFSA membrane with organic/inorganic additives, and development of alternative polymeric composites.
To address the methanol permeability in PFSA membranes, several attempts have been reported in the literature by dispersing variety of inorganic additives like silica, zirconia, metal oxides and zeolites to form Nafion® composites [8-11]. These composite membranes show restricted methanol permeability with a compromise of proton conductivity in DMFCs. On the other hand, many organic additives have been explored to form composite, blend and cross-linked membranes of Nafion® [12-15] which has shown better proton conductivity. Dispersing carbon nanomaterials in Nafion is a recent trend to improve the physic-chemical properties of Nafion. This is given in . section 1, page 4, and paragraph 2-4 of the revised manuscript and highlighted in yellow.
Comment 5: You never talk about the life of its compounds, can you specify this point in your review?
Authors response: Stability aspects of the composite membranes are discussed where ever appropriate.
The durability of the membranes assessed by static (OCV) and dynamic (load of 200mA/cm2) for 100h with 5M methanol solution. In both the condition no obvious drop in performance is observed, however further investigation on life time of these composite membranes are essential to prove their suitability as PEMs for DMFCs.
The lifetime analysis of the composites also performed by recording OCV as a function of time (up to 4days) wherein Pt-Ru-CNTs composites shows improved OCV as function of time on the other hand neat SPAS membrane shows drop in OCV due to severe methanol crossover.
The stability of SPI-SPSGO composite membrane is evaluated in a single-cell by measuring the current density of the cell operating at 0.6V. However, the test has been conducted only for 70h and observed marginal performance drop long term stability (life-time) of this membrane should be adequately studied for there is as PEMs in DMFCs.
The stability test was carried out for SDBS-HGO paper under a constant load of 50mA cm-2 where in SDBS-HGO shows higher stability than Nafion-112 and thus better lifetime can be expected for this type of GO papers.
The stability of SPPEK-SGNF membrane is also examined in terms of methanol crossover current density using linear sweep voltammetry (LSV). Methanol crossover current density through the MEAs comprising pristine SPPEK and SPPEK-SGNF was measured after 50 h OCV operation and compared the results with initial methanol crossover current density values measured in similar conditions. It is found that pristine SPPEK shows two-fold increment in crossover current density whereas the composite membrane show marginal increment in crossover current density suggesting its methanol blocking characteristic and stability under DMFC operating conditions.
This is given in (section 2.1 page 12 paragraph 1, section 2.2, page 12 paragraph 4, section 3.3, page 19, paragraph 3, section 3.6 page 23 paragraph 1, section 4,page 27, paragraph 1) in the revised manuscript. Nevertheless, lifetime analysis of these composites is not adequately studied in the literature and is the future subject of study. All the changes are highlighted in yellow in the revised manuscript.
Reviewer 4 Report
The authors report a review on carbon nanocomposite membrane electrolytes for direct methanol fuel cells-A concise review. Indeed, the review is important in the field of carbon nanocomposite membrane electrolytes. Before publication of the manuscript in nanomaterials, the following concerns should be addressed.
There are some typo errors throughout the manuscript. Authors should carefully correct. Authors should include the “Future perspective” as separate subtitle before conclusive remarks. It should contain future direction on carbon fillers, carbon nanocomposite electrolytes for DMFC. Figure 8 seems to have low resolution. Please improve the resolution. Figure 9 indicate about the different functionalization methods for GO, but the explanation bout how such GOs can improve the membrane performance is not sufficient. So, please give more explanation. The literature review is not sufficient, the authors should include some related articles: doi.org/10.1016/j.jechem.2018.02.020; doi.org/10.1016/j.compositesb.2018.11.084; doi.org/10.1021/acssuschemeng.9b01757; doi.org/10.1002/er.4494.Author Response
Comments Reviewer 4:
Comment1: The authors report a review on carbon nanocomposite membrane electrolytes for direct methanol fuel cells-A concise review. Indeed, the review is important in the field of carbon nanocomposite membrane electrolytes. Before publication of the manuscript in nanomaterials, the following concerns should be addressed.
Authors response: We thank the reviewer for spending his/her valuable time on reviewing the manuscript and his positive remarks on the manuscript.
Comment 2: There are some typo errors throughout the manuscript. Authors should carefully correct.
Authors response: Authors apologize for the typo error. All the typo errors are corrected now in the revised manuscript.
Comment 4: Authors should include the “Future perspective” as separate subtitle before conclusive remarks. It should contain future direction on carbon fillers, carbon nanocomposite electrolytes for DMFC.
Authors response: As suggested by the reviewer future prospective is separated for the conclusion section and future research directions of the carbon nano-fillers are highlighted.
Future prospects
Carbon nanomaterials are proved to be promising reinforcing materials in PEMs for fuel cell technology in which carbon composites are recently emerged as promising materials. By the proper tuning of their properties and adequate use of carbon fillers in PEMs can provide the solutions for most of the current limitations in PEMs and helps in advancing the fuel cell technology to be available for day-to-day utilization. On the other hand, one of the key issue lifetime of these carbon-nanocomposites are not adequately addressed in the literature so far, stability tests are hardly carried out up to few hundred hours, but for commercial DMFC the expected lifetime is much higher. Hence, future studies need to clarify this point to progress in carbon-polymer nanocomposite PEMs for DMFCs. In addition, carbon nanofillers are promising materials in many other technologies such as water filtration, desalination, and flow batteries. However, large scale, cost-effective production are the key challenges in addition to their dispersion. Hence adequate investigations should be carried out in terms of low-cost production and improved dispersion to extend their uses to numerous technologies.
This is given in Section 6, page 27 of the revised manuscript and highlighted in yellow.
Comment 4: Figure 8 seems to have low resolution. Please improve the resolution.
Authors response: The resolution of figure 8 is improved in the revised manuscript.
Comment 5: Figure 9 indicate about the different functionalization methods for GO, but the explanation about how such GOs can improve the membrane performance is not sufficient. So, please give more explanation.
Authors response: As asked by reviewer explanation about functionalized GOs is given in terms of improved membrane properties.
GO was functionalized with SDBS wherein it is adsorbed in the surface of GO through π-π and hydrophobic interactions. Incorporation of SDBS-GO in SPEEK reduces the dimension of the ionic cluster from 1.96 to 1.76 nm thus mitigating the methanol permeability. The reduction in ionic cluster sis mainly associated increased interactions benzene rings in SDBS and hydrophobic backbone and sulfonic acid groups of SDBS and hydrophilic clusters of SPEEK.
Another method of sulfonating GO is by using propane sultone. Unlike in SDBS-GO, in this case, there is a chemical bond between GO and sulfonic acid. By using this sulfonated GO as an additive in SPEEK there are certain differences in properties of the composite membranes in comparison with SPEEK-SDBS-GO attributed to the difference in the functionalization method.
Sulfanilic acid is treated with hydrochloric acid and sodium nitrate to form the 4-benzenediazonim salt. The formed precursor is then treated with GO to form sulfonic acid functionalized GO through new C-C bond between sp2 carbon of GO and sp2 carbon of benzene sulfonic acid. When this S-GO is used as an additive in SPEEK, as reported in the literature, SGO improves the homogeneous dispersion of GO in SPEEK due to enhanced compatibility between sulfonated groups of GO and SPEEK.
Unlike previous reports, GO was functionalized with amino acid wherein proton transport occurs via amine groups and carboxylic acid groups. On the other hand, amino acid forms electrostatic interactions with water molecules and held between –NH2+– and –COO- groups. These bridged water molecules form a hydration layer in the composite membrane acting as a vehicle for proton transport.
This is given in Section 3.2, page18, paragraph 2 and 3, page 19, paragraph 1 and 2 of the revised manuscript and highlighted in yellow.
Comment 6: The literature review is not sufficient, the authors should include some related articles: doi.org/10.1016/j.jechem.2018.02.020; doi.org/10.1016/j.compositesb.2018.11.084; doi.org/10.1021/acssuschemeng.9b01757; doi.org/10.1002/er.4494.
Authors response: As suggested by the reviewer literature related to carbon composite are added in the revised manuscript. (ref. 31, 32, 60, 125).
Round 2
Reviewer 3 Report
dear author
I thank you for taking into account my remarks, comments for the revision of your article. I have no additional comment on your work
best regards
Reiewer
Author Response
Authors response: We thank you for your valuable comments and thanks for appreciating our work.